The first occurrence of machimosaurid crocodylomorphs from the Oxfordian of south-central Poland provides new insights into the distribution of macrophagous teleosauroids

http://orcid.org/0000-0001-7036-0368 Weryński Łukasz 1 lukaszwerynski@doctoral.uj.edu.pl
Błażejowski Błazej 2
http://orcid.org/0000-0001-5108-8493 Szczygielski Tomasz 2
http://orcid.org/0000-0002-7263-6505 Young Mark T. 3
1 Institute of Geological Sciences, Jagiellonian University in Kraków , Kraków, Małopolska , Poland
2 Institute of Paleobiology, Polish Academy of Sciences , Warsaw, Masovian Voivodeship , Poland
3 Grant Institute, The King’s Buildings, University of Edinburgh , Edinburgh, School of GeoSciences , United Kingdom
Anquetin Jérémy
Electronic publication date: 2024 Mar 28
Publication date: 2024
Volume: 12
Electronic Location ID: e17153
Received 2024 Feb 1; Accepted 2024 Mar 4
Copyright: © 2024 Weryński et al.
Copyright year: 2024
Copyright holder: Weryński et al.
License: This is an open access article distributed under the terms of the Creative Commons Attribution License, which permits unrestricted use, distribution, reproduction and adaptation in any medium and for any purpose provided that it is properly attributed. For attribution, the original author(s), title, publication source (PeerJ) and either DOI or URL of the article must be cited.
License URL: https://creativecommons.org/licenses/by/4.0/

Keywords: Crocodylomorphs, Machimosaurid, Macrophagous, Rostrum, Teeth, Jurassic, Oxfordian, South-central Poland

Funding: Polish National Science Center 2020/39/B/ST10/01489 Priority Research Area (WSPR.WGiG.1.5.2022.2) under the Strategic Programme Excellence Initiative at Jagiellonian University This research was supported by the Polish National Science Center Grant no. 2020/39/B/ST10/01489 and Priority Research Area (WSPR.WGiG.1.5.2022.2) under the Strategic Programme Excellence Initiative at Jagiellonian University. The funders had no role in study design, data collection and analysis, decision to publish, or preparation of the manuscript.

==============================
Teleosauroid thalattosuchians were a clade of semi-aquatic crocodylomorphs that achieved a broad geographic distribution during the Mesozoic. While their fossils are well documented in Western European strata, our understanding of teleosauroids (and thalattosuchians in general) is notably poorer in Central-Eastern Europe, and from Poland in particular. Herein, we redescribe a teleosauroid rostrum (MZ VIII Vr-72) from middle Oxfordian strata of Załęcze Wielkie, in south-central Poland. Until now, the specimen has been largely encased in a block of limestone. After preparation, its rostral and dental morphology could be evaluated, showing the specimen to be a non-machimosaurin machimosaurid, similar in morphology to taxa Neosteneosaurus edwardsi and Proexochokefalos heberti. The well-preserved teeth enable us to study the specimen feeding ecology through the means of comparing its teeth to other teleosauroids through PCoA analysis. Comparisons with inferred closely related taxa suggest that the referred specimen was a macrophagous generalist. Notably, MZ VIII Vr-72 displays a prominent pathological distortion of the anterior rostrum, in the form of lateral bending. The pathology affects the nasal passage and tooth size and position, and is fully healed, indicating that, despite its macrophagous diet, it did not prevent the individual from food acquisition.

Introduction

Jurassic seas hosted a wide array of large macro-predatory vertebrates, which were diverse both taxonomically and morphologically, not unlike in current marine ecosystems. Unlike modern ecosystems, however, marine reptiles filled many of the large-bodied predatory niches, including those at the top of the food chain (e.g. Massare, 1987, 1988; Martill et al., 1994; Buchy, 2010; Foffa et al., 2018). One significant marine reptile group was Thalattosuchia, a diverse crocodylomorph clade known from the Jurassic and Early Cretaceous. Thalattosuchia is comprised of two subclades: the Metriorhynchoidea, which transitioned into being exclusively aquatic (e.g. Fraas, 1902; Andrews, 1913; Piveteau, 1928; Buffetaut, Termier & Termier, 1981; Buffetaut, 1981; Young et al., 2010; Wilberg, 2015; Ősi et al., 2018; Cowgill et al., 2023), and the Teleosauroidea, a semi-aquatic subclade long considered to be marine analogues of the extant gharial (Andrews, 1909, 1913; Buffetaut, 1981; Hua, 1999). During the Jurassic, teleosauroids achieved a broad geographic distribution, inhabiting freshwater, brackish, and coastal marine ecosystems (Andrews, 1913; Buffetaut, 1981; Buffetaut, Termier & Termier, 1981; Hua & Buffetaut, 1997; Vignaud, 1997; Hua, 1999; Young et al., 2014a; Martin et al., 2019, 2016; Johnson et al., 2017, 2018, 2019; Johnson, Young & Brusatte, 2020a, 2020b; Foffa et al., 2018; Wilberg, Turner & Brochu, 2019), possibly also surviving to the Early Cretaceous (Fanti et al., 2016; Cortés et al., 2019).

Based on our knowledge from Western Europe, the Oxfordian is a key interval in teleosauroid biodiversity. The Callovian teleosauroid fauna was dominated by non-machimosaurin machimosaurids, while by the Kimmeridgian, aeolodontin teleosaurids and machimosaurin machimosaurids were the most abundant (Johnson, Young & Brusatte, 2020b).

The fossil record of the crocodylomorphs in Poland is currently quite limited, especially for well-preserved specimens (Zatoń, 2007). Most of the teleosauroid specimens found in Poland have either been assigned to the genus Machimosaurus or to ‘Steneosaurus’ (Deecke, 1907; Dzik, 1992, 1997, 2003, 2011; Hoffmann, 2005; Hoffmann & Bickelmann, 2008; Madzia, Szczygielski & Wolniewicz, 2021; see Table 1). The majority of these fossils, belonging to Thalattosuchia, have been found in the region of Pomerania (northern Poland), including the notable historical records of Schmidt (1905) and Krebs (1967). Dzik (1992) also noted the presence of a teleosauroid skull fragment in the collection of the Greifswald Museum, with collection number GG303-30. The Pomeranian sites at which Teleosauroidea fossil material have been documented include Czarnogłowy, Kłęby, and Wrzosowo (see Table 1).

Table 1 Notable occurrences of Teleosauroidea in fossil record of Poland (modified from Madzia, Szczygielski & Wolniewicz, 2021).

Locality	Age	Material	Specimen collection number	Original taxonomic referral	Revised taxonomic referral	
Czarnogłowy (Zarnglaff)	K	Teeth	GG303-32, GG303-33	Steneosaurus jugleri (Deecke, 1907; Hoffmann, 2005; Hoffmann & Bickelmann, 2008)	Not revised	
Teeth, vertebrae	GG303-31	Steneosaurus sp. (Hoffmann, 2005; Hoffmann & Bickelmann, 2008)	Not revised	
Osteoderm	GG303-41	Teleosauridae indet. (Hoffmann, 2005)	Steneosaurus sp. (Hoffmann & Bickelmann, 2008)	
Vertebral centra	GG303-43, GG303-44, GG303-45, GG303-46, GG303-47C	Thalattosuchia indet. (Hoffmann, 2005)	Steneosaurus sp. (Hoffmann & Bickelmann, 2008)	
Mandible fragment	GG303-30	Machimosaurus sp. (Dzik, 1992)	Machimosaurus hugii (Dzik, 1997)	
Steneosaurus sp. (Hoffmann, 2005; Hoffmann & Bickelmann, 2008; Young et al., 2014a)	
Teeth, mandible fragment, vertebrae	GG303-34, GG303-35, GG303-36, GG303-40	Machimosaurus hugii (Dzik, 1992, 1997, 2003)	Machimosaurus sp. (Hoffmann, 2005)	
Machimosaurus hugii (Hoffmann & Bickelmann, 2008; Dzik, 2011)	
Teeth	GPIT/RE/328, GPIT/RE/9280, GPIT/RE/9281	Machimosaurus cf. buffetauti (Young et al., 2014a)	Not revised	
Unspecified remains	not given	Machimosaurus sp. (Schmidt, 1905)	Not revised	
Kłęby (Klemmen)	O	Teeth	GG303-37	Machimosaurus sp. (Hoffmann, 2005, 2007)	Not revised	
Wrzosowo	K	Teeth	GG303-38, GG303-39	Ichthyosaurus? sp. (Sadebeck, 1865)	Machimosaurus? sp. nov. (Dames, 1888)	
Machimosaurus sp. (Schmidt, 1905; Deecke, 1907; Hoffmann, 2005)	
Machimosaurus hugii (Krebs, 1967; Hoffmann & Bickelmann, 2008)	
Załęcze Wielkie	O	Mandible and upper jaw fragment	MZ VIII Vr-72	Peloneustes sp. (Maryańska, 1972)	Teleosauridae indet. (Ketchum & Benson, 2011; Tyborowski & Błażejowski, 2019b)	

The first well-documented occurrence of the Thalattosuchia in south-central Poland is an incomplete teleosauroid rostrum. This specimen (MZ VIII Vr-72) was discovered in the Kraków-Wieluń Upland, near the Załęcze Wielkie village, and was described by Maryańska (1972). While she interpreted the specimen as a pliosaurid (of the genus Peloneustes), the specimen had not been fully prepared at that time. The other known occurrences of thalattosuchians from south and central Poland are of metriorhynchids (Madzia, Szczygielski & Wolniewicz, 2021). One, described by Zatoń (2007), is a metriorhynchid tooth found in the Ogrodzieniec quarry, subsequently identified as cf. Tyrannoneustes sp. (Young et al., 2013a). The second occurrence, from central Poland, an isolated tooth from Inowrocław, is either Tyrannoneustes or a related taxon (Gallinek, 1896; Young et al., 2013a). There are also occurrences of thalattosuchians from margin of Holy Cross Mountains (Madzia, Szczygielski & Wolniewicz, 2021).

Herein, we redescribed the partial rostrum (MZ VIII Vr-72). The preserved dentition and cross sections of the rostrum of MZ VIII Vr-72 both in vertical and horizontal planes, along with CT scans, allowed us to investigate the distinctive characteristics of that teleosauroid specimen with respect to its rostral and dental morphology.

Specimen background

Geological background

Załęcze Wielkie (51°5′13″ N, 18°41′9″ E; Fig. 1) is located in the Pątnów community, Wieluń county, Łódzkie voivoidship, in the northernmost part of the region called the Kraków-Wieluń Upland. The Kraków-Wieluń Upland, or more colloquially Polish Jura (Gradziński et al., 2011), is a physiographic region of south-central Poland, stretching from Kraków to Wieluń, being a part of the larger Kraków-Silesian Upland. It is divided into the Kraków, Częstochowa and Wieluń Uplands (Matyszkiewicz, Krajewski & Żaba, 2006). Geologically, the region is a part of the Kraków-Silesian Monocline. It is mainly known for its extensive surface presence of Upper Jurassic rocks, building picturesque rock formations (Jezierski, 2008; Gradziński et al., 2011) and strong karst phenomena that occur in the region, which are responsible for the formation of many caves (Urban, 2004, Gradziński et al., 2011). The quarries of the Polish Jura are one of the most important sources of limestone in Poland (Wierzbowski, Matyja & Ślusarczyk-Radwan, 1983).

Figure 1 The location of Załęcze Wielkie, the acquisition site of the specimen MZ VIII Vr-72 (tan outline of a machimosaurid based on the paleoart of Stanisław Kugler from Fig. 10).

Figure 10 The paleoart depicting the speculative appearance of the MZ VIII Vr-72 species.

Reconstruction of typical member of species (A) and particular specimen with distorted rostrum (B) by Stanisław Kugler.

The locality of Załęcze Wielkie lies in the north-western part of the Wieluń Upland, at the Warta river. The precise historical position of the section the described fossil came from is, however, difficult to locate. A heavily quarried area has been placed between Załęcze Wielkie and Załęcze Małe, mostly at the hamlets of Zamłynie and Troniny, but the quarries are currently abandoned and overgrown. They revealed various limestones of the Częstochowa Sponge Limestone Formation (Wierzbowski, 1978, 1980), both the bedded and massive limestones mostly of the middle Oxfordian age (Perisphinctes plicatilis to Gregoryceras transversarium ammonite Zones). The lithology of the matrix surrounding the specimen suggests that it was collected from the bedded limestones–possibly the so called ‘grey limestones’ which commonly occur in the area from the uppermost Lower to the lower part of the Middle Oxfordian (Perisphinctes plicatilis ammonite Zone). These ‘grey’ limestones appear to be formed in slightly anoxic environment (A. Wierzbowski, 2023, personal communication), and this interpretation would partly explain the presence of a well, three-dimensionally preserved fossil of a large animal from this horizon.

Historical background

The examined rostrum (MZ VIII Vr-72 or ‘Załęcze Wielkie specimen’) is part of the collection in the Museum of the Earth (MZ), Warsaw. It was initially described and illustrated by Maryańska (1972: pl. 1) who identified it as a small pliosaurid (Peloneustes sp.). However, Ketchum & Benson (2011) subsequently re-identified the Załęcze Wielkie specimen as a teleosaurid crocodylomorph.

Tyborowski & Błażejowski (2019a, 2019b) described a marine vertebrate assemblage from the Upper Jurassic (Kimmeridgian) limestone beds of Krzyżanowice in the NE margin of the Holy Cross Mountains in Poland, a locality historically revealed to yield vertebrate (turtle) macrofossils by Młynarski & Borsuk-Białynicka (1968). The Załęcze Wielkie specimen (original MZ VIII Vr-72 of Maryańska, 1972) ended up sharing its catalogue number with another specimen of an uncertain age and provenance, at the time considered to represent the Krzyżanowice assemblage (see discussion in Madzia, Szczygielski & Wolniewicz, 2021). This confusion combined with a lack of thorough investigation into the museum’s archival collections, led (Tyborowski & Błażejowski 2019b: fig. 6) to mistakenly include the Załęcze Wielkie specimen in the Krzyżanowice fossil collection. Fortunately, this error was spotted and corrected by Madzia, Szczygielski & Wolniewicz (2021). The same catalogue number (MZ VIII Vr-72) was also historically assigned to a lepidosauromorph dentary from the Olenekian of Czatkowice (Borsuk-Białynicka et al., 1999: fig. 5A).

METHODS AND TERMINOLOGY

Institutional abbreviations

GG, Institut für Geographie und Geologie, Ernst-Moritz-Arndt-Universität Greifswald, Germany; GIUS, Institute of Earth Sciences, Faculty of Natural Sciences, University of Silesia, Katowice, Poland; GPIT, Palaeontological Collection of the University of Tübingen, Germany MZ, Polish Academy of Sciences Museum of the Earth, Warsaw.

Fossil preparation and micro-CT methodology

The preparation of the rostrum (Figs. 2–6) was conducted using an air scribe. The preparation of the whole specimen proved to be challenging, due to the very compact nature of limestone concretion which encased the specimen, and as such, the bone was exposed to such degree to not damage the specimen. The prepared, separated three sections (labelled F1, F2 and F3) were then scanned using X-ray microtomography (XMT or micro-CT) at the Institute of Earth Sciences of the University of Silesia (GIUS) to acquire a 3D image of the rostrum and teeth surface together with their internal structures. A number of virtual cross-sections was generated to show internal structure of the specimens (Fig. 7). Tomographic data after digital processing enable the construction of isosurface-based and volume-based 3-D (a ‘virtual fossils’), which can be manipulated, dissected or measured interactively. The resulting images are similar to traditional ones obtained by destructive slicing, and the resolution here is limited by voxel size of computed model, reaching 22.4 × 22.4 × 22.4 μm. The reconstructed computed tomography (CT) data were converted into TIFF image stacks that were subsequently imported and segmented in VGStudio MAX version 3.0 (Volume Graphics Inc., Heidelberg, Baden-Wurttemberg, Germany). Additionally, for the purposes of the photography, the teeth of the specimen were coated with sublimed ammonium chloride to highlight the structure of the enamel surface.

Figure 2 The overview of the MZ VIII Vr-72.

(A) The whole rostrum as placed in the limestone block in lateral. (B) The closeup of the left section of the rostrum with visible structure of the maxillary teeth and the cross-section through the left maxilla and dentary. (C) Overview of the whole left section (as originally described by Maryańska (1972)).

Figure 3 The overview of the premaxilla.

Note the dorsal location of the external nares (A), which are placed on strongly distorted premaxilla, with substantial reception pits on the bone surface. In ventral view, the premaxilla is hardly exposed (B). In contrast to premaxilla and anterior maxilla, the whole dentaries (C) appear to be non-distorted.

Figure 4 The F3 section of the MZ VIII Vr-72 specimens’ rostrum.

Photos (A, B and D) documenting: (A) dorsal view; (B) visible bone surface with numerous and well-developed neurovascular foramina; (D) longitudinal cross section with observable distortion of the most anterior maxillae and premaxilla along with the right dentary. The CT scans (C and E): (C) documenting the fragment F3 of the upper jaw in ventral view, highlighting the distortion of the anteriormost rostrum and with visible teeth and alveoli of the premaxilla and maxilla; (E) left lateral view of the premaxilla fragment and the anterior maxillary with strongly visible superficial reception pits and inflection of the anterior rostrum.

Figure 5 The F2, central section of rostrum.

Photos (A, C, E, F, H–J) documenting: (A) dorsal view; (C) ventral view; (E) right lateral view; (F) dental structure of the MZ VIII Vr-72 with almost completely preserved tooth and impaction process captured in fossil; (H and J) vertical cross sections through dentaries and maxillae; (I) longitudinal cross section through dentaries and maxillae. CT scans of the F2 (B, D and G). (B) scan of the upper jaw; (D) scan of the mandible; (G) scan of the mandible in dorsal view.

Figure 6 F1, posteriormost section of rostrum of MZ VIII Vr-72.

Photos (A, C, E, F, H–J) documenting: (A) dorsal view (with visible splinter of nasal); (C) ventral view; (E) right lateral view; (F) dental characteristics of the MZ VIII Vr-72: teeth with conspicuous pattern of apicobasal ridges, robust in form, with mid-crown curvature and apical anastomosing pattern, which is especially visible on the first tooth to the left; (H and J) vertical cross sections through dentaries and maxillae; (I) longitudinal section through maxillaries and dentaries. CT scans of the F1 (B, D and G): (B) scan of upper jaw; (D) scan of mandible; (G) scan of mandible in lateral view documenting visible slippage of dentary teeth and neurovascular foramina parallel to tooth row.

Figure 7 MZ VIII Vr-72 maxilla in XMT scans.

(A–C) Documenting the variation in the shape of the nasal cavity, dental alveoli structure and dorsal alveolar canals through the length of the rostrum. Note the increasing disfiguration of nasal cavity and dorsal alveolar canals at the anteriormost maxilla and premaxilla.

Tooth description and morphometric ordination analysis

The terminology of tooth orientation is based on teleosauroid teeth studies (e.g. Massare, 1987; Vignaud, 1997; Young et al., 2013b, 2014b) with the following terms used: apical–towards the apex of the tooth crown; basal–towards the tooth crown base; mid-crown–approximately centrally between the crown apex and base; mesial–towards the anterior direction of the animal’s mouth; distal–inwards into the animal’s mouth; labial–towards the animal’s lips; lingual–towards the tongue.

Three of the selected teeth of the specimen, which were completely erupted and characterized by intact crowns exposed in the middle section of the rostrum, were measured using an electronic calliper to acquire continuous characters for the Principal Coordinates Analysis (PCoA). The measured values of the three teeth could be averaged for the needs of the analysis. Together with observable discrete characters, these parameters were used in the PCoA. The analyses were conducted in PAST 4.13 (Hammer, Harper & Ryan, 2001) to allow the placement of the Załęcze Wielkie specimen in the morphospace of Johnson’s et al. (2022a) data matrix of teleosauroid tooth morphotype variation. For the needs of the PCoA analysis, continuous characters were Z-transformed and Gower similarity index (Gower, 1971) was used, as it is well-suited for utilizing datasets containing both continuous and discrete characters (Madzia, Szczygielski & Wolniewicz, 2021), with transformation exponent of c = 6.

Classification

The classification of the described specimen is based on recommendations of International Commission on Zoological Nomenclature (ICZN).

Supplementary material and data availability

The supplementary data attached in Supplementals 1–3 are measurements and observations of teleosauroid teeth morphologies, based on Johnson et al. (2022a) data used in PCoA analysis. The measurements of the teeth and the characters that have been observed in MZ VIII Vr-72 are available in Supplemental Material 1. The modified data matrix from Johnson et al. (2022a) is present at Supplemental Material 2 and additional results of analyses, with eigenvalues and different coordinate variants can be found in Supplemental Material 3. The virtual scans of specimen MZ VIII Vr-72 rostrum are available at MorphoSource:

The F1 section: https://doi.org/10.17602/M2/M609339 (CT images series) https://doi.org/10.17602/M2/M600506 (3D mesh); The F2 section: https://doi.org/10.17602/M2/M609720 (CT images series) https://doi.org/10.17602/M2/M600521 (3D mesh); The F3 section: (CT images series) https://doi.org/10.17602/M2/M609868 (CT image series); https://doi.org/10.17602/M2/M600532 (3D mesh); Full composite of F1, F2 and F3 3D meshes: https://doi.org/10.17602/M2/M609361.

Specimen description

Systematic palaeontology

CROCODYLOMORPHA Hay (1930) (sensu Nesbitt, 2011)

THALATTOSUCHIA Fraas (1901) (sensu Young et al., 2024)

TELEOSAUROIDEA Geoffroy Saint-Hilaire (1831) (sensu Johnson, Young & Brusatte, 2020b)

MACHIMOSAURIDAE Jouve et al. (2016) (sensu Johnson, Young & Brusatte, 2020b)

MACHIMOSAURINAE Johnson, Young & Brusatte (2020b)

MACHIMOSAURINAE gen. et sp. indet. (Figs. 2–6)

Material: MZ VIII Vr-72, an incomplete rostrum (premaxilla, maxilla, dentaries, and a small fragment of the nasals), with well-preserved in situ dentition.

Locality and horizon: Załęcze Wielkie, south-central Poland; probably from the Perisphinctes plicatilis ammonite Zone of the middle Oxfordian Częstochowa Limestone Sponge Formation, bedded ‘grey’ limestones.

Remarks: The original description of the specimen by Maryańska (1972) is very succinct and somewhat difficult to follow, especially as the author did not clearly state whether the described characters unambiguously pertain to its original anatomy, result from its imperfect preservation, or from the incomplete preparation. Moreover, the photographic documentation of those characters is very limited. Thanks to additional preparation and CT data, some of the characters can be re-evaluated.

Preservation

The Załęcze Wielkie specimen (MZ VIII Vr-72) is preserved in four pieces: one block (Fig. 2) embedding small parts of the left maxilla, the left premaxilla and fragment of right premaxilla, and most of the anterior part of the left dentary (Fig. 3) with a minor remainder of the right dentary; and three consecutive fragments (F3, F2 and F1, Figs. 4–6) including most of the preserved maxillae, marginal fragment of right premaxilla, parts of the dentaries (mostly right). The original description by Maryańska (1972) correctly noted that the presented rostrum fragment covers a portion of the symphyseal region of the mandible. However, the mandible seems to be mostly broken obliquely through the right dentary in the ventral part and through the left dentary more dorsally, not along the symphysis, with only a small area with a lamellar structure (representing the symphysis) exposed in the posterior part. The bones remaining in the block are mostly exposed along their break surfaces so their informativeness is limited. Overall, aside of the intermaxillary suture, the bone sutures are poorly defined.

The preserved, separated right and medial part of the rostrum is 33.5 cm long. In its posterior-most part it includes a short portion of both transversely complete maxillae, revealing the total width of the snout at that level to be 5.6 cm (Fig. 6). This separated rostrum fragment is fractured into three consecutive sections, all of which contain fragments of maxillae and dentaries. There is an observable distortion of the anterior portion of the rostrum along the midline axis, as the anterior maxilla is significantly and sharply deflected to the left from its original position. A similar, albeit gentler condition localized more distally in the rostrum, can be observed in the holotype of Mycterosuchus nasatus from England (NHMUK PV R 2617; Andrews, 1913).

The remaining part of the rostrum represents a parasagittal section of the snout through the left maxilla and dentaries with exposed dental alveoli and partially preserved teeth. This part of the rostrum is longer than the detached portion, 37 cm long, and includes more of the posterior part of the dentaries (Fig. 2).

Premaxilla

Both premaxillae are present in the specimen. The marginal fragment of the right premaxilla constitutes a section of the anterior-most piece separated from the block and is split obliquely from the rest of the right premaxilla and the left premaxilla (Figs. 3, 4). The premaxillary region of the rostrum is disfigured by a prominent and relatively sharp bend of over 90° (Fig. 3), as evidenced by the direction of the nasal passage. In the CT scan, the ventral view of the right premaxilla is exposed, with visible right fourth premaxillary tooth along with a fragment of the third dental alveolus (Figs. 4, 7). The precise tooth morphology of the premaxilla is not observable, as only a small area of the bone is exposed in ventral view. The external nares are well-visible in anterior and dorsal views (Fig. 3). The edge of the posterior portion of nares takes the form of a shallow, smooth depression separated from the external surface of the bone by a marked ridge (Fig. 3). There is a visible lateral expansion of the premaxilla relative to maxilla, but the prominence of this trait is less evident when compared to other teleosauroid specimens due to the sharp distortion of anterior rostrum. There appears to be only moderate constriction of the premaxilla relatively to maxilla, and undeformed bone was probably horizontally straight. The premaxillary-maxillary suture is not traceable either in dorsal or in ventral view. The surface of the premaxilla is covered by prominent reception pits (Fig. 3A)

Maxilla

In Teleosauroidea, the maxilla forms the substantial part of the rostrum and is one of the largest bones on the skull (e.g. see Andrews, 1913; Johnson, Young & Brusatte, 2020b). In MZ VIII Vr-72 (Figs. 2, 4–6), the maxilla is rather simple in form, slightly tapering toward the anterior end of the snout, with an ovaloid cross-section that is wider than high. The bone is strongly elongated and can be considered to be intermediate in robustness when compared to most teleosauroid genera (see Johnson, Young & Brusatte, 2020b), based on the measured proportion of width to length and observed general proportions. The anteriormost section of the maxilla is also disfigured by the lateral bend to the left as a continuation of the premaxillary inflection. The surface of the bone exhibits a distinct superficial bone ornamentation consisting mostly of narrow, longitudinally or posteromedially inclined grooves. Most of the lateral surface of the left maxilla is separated from the rest of the bone and remains attached to the rock block, exposing deep dental alveoli that extend nearly to the dorsal surface of the bone. The tooth sockets contain a set of teeth at various stages of growth, some of which have exposed pulp cavity.

In the dorsoventral aspect, the exposed lateral maxillary margin exhibits small undulations caused by tooth socketing, but aside of that the lateral edges of bone are generally straight and parallel in dorsal view. The lateral margins of the bone are also perforated by distinct neurovascular foramina that are lined linearly parallel to the teeth line. In addition to the foramina, the surface of the bone is dotted with small reception pits, which are most numerous on the lateral surfaces and in the anterior region of the preserved bone adjacent to the premaxilla. Mesially, both maxillae are parted by a clearly visible midline suture. The precise tooth count for the whole bone is hard to estimate due to its incompleteness, but in the fragment described herein there are 18 teeth and alveoli preserved on the left maxillary, as has been originally noted by Maryańska (1972), while the right maxillary has 19 preserved alveoli. The observed alveoli appear to be very prominent and comparatively large.

The nasal cavity is spacious, and visible in the natural cross sections and images based on XMT scans (Fig. 7). It is an inverted heart-shaped and subdivided into three sections: the largest, ventral nasal passage, immediately dorsal to the secondary bony palate, has two sub-chambers only partially divided ventrally by a mesial bony ridge and becomes flatter oval towards the anterior end of the snout. Throughout the preserved part of the snout, the septal ridge is clearly defined but relatively low and there is no evidence of a septal sulcus.

There are two paired cavities dorsal to the nasal cavity. They are irregularly shaped and separated from the nasal cavity and each other by thin bony septa. These are most likely the dorsal alveolar canals (Pierce, Williams & Benson, 2017; Bowman et al., 2022). The dental alveoli contain the tooth roots, and are located lateral and dorsolateral to the nasal cavity. The rostrum does not preserve any pneumatic infiltrations of the bony palate or maxillary rostrum, which is consistent with Thalattosuchia (Witmer, 1997; Jouve, 2009; Fernández & Herrera, 2009; Herrera, Fernández & Gasparini, 2013; Pierce, Williams & Benson, 2017; Cowgill et al., 2022). There are also no palatal (maxilla-palatine) grooves preserved, which is consistent with Middle and Late Jurassic teleosauroids (see Young et al., 2023).

Nasal

Very little of the nasals are present, as the preserved rostrum fragment is almost entirely composed of the maxilla. However, what appears to be a small, narrow (11 × 1 mm) part of the nasals is wedged between the posterior ends of the maxillae (Fig. 6). The preserved nasals are triangular in shape, as only the tip of the anterior process is present. There is no evidence of a midline suture, although whether this is real or artefactual is unknown.

Dentary

The dentaries are the only bones of the mandible that are preserved. Both dentaries are partially preserved but they are mostly embedded in the block of rock matrix and thus largely inaccessible. The detached and prepared fragments of the rostrum include parts of the right dentary. The anterior-most portion of the dentaries was originally present at the time of the recovery, as indicated by the remaining negative space and a natural mold of its dorsal surface (including the symphysis and alveoli), but this section is now missing. The ventral surface of the symphysis is mostly embedded in the matrix but the interdental suture is partly exposed in the posterior part. In contrast to the premaxilla and anterior maxilla, the dentaries appear to be straight, without any disfiguration.

The separate sections of dentaries are oval in the cross-section, almost as tall as wide, and the whole symphyseal region consisting of dentaries itself is rather narrow and elongated, being noticeably narrower than upper jaw. Its vertical dimensions are considerate, being similar in depth to the upper jaw. The surface of the dentary is moderately rough and dotted with neurovascular foramina along the ventrolateral edge parallel to the tooth row (Figs. 5, 6), and the ventral surface of the bone exhibits a moderately wavy, irregular surface with roughly longitudinal superficial ornamentation.

The preserved part of the detached, right dentary has 14 dental alveoli visible (the distal-most alveolus is only partially preserved). There is an observable displacement of the teeth from their sockets in the dentary, exhibited as an exposure of a significant portion of the roots in preserved teeth and their removal from their original position within the teeth sockets. The left part of the dentary, which is a part of the large block, has exposed ventromedial parts of the dental alveoli, which allow one to easily observe the depth of the root within bone. In the longitudinal cross section (Figs. 2, 4–6), there is a small discernible cavity observable within the bone cross-section.

Dental characteristics

The tooth crowns are robust and slightly recurved labiolingually. The dentition is generally uniform along the whole length of the jaw. This is expected, as teleosauroids have a generally homodont dental morphology (Johnson et al., 2022a, Machimosaurus however exhibits stronger degree of heterodonty, see Vignaud, 1995; Hua, 1999). However, like other teleosauroids, the posterior teeth are slightly more robust than the anterior dentition, with slightly wider and shorter crowns. The average apicobasal crown height/basal crown diameter index is 2.17 based on three measured dentary and maxillary teeth which have completely preserved crowns. The apices of the teeth are pointed but robust, gently rounded. The root forms more than two-thirds of total tooth length. The roots are curved distolingually, and both in the cross sections of the rostrum and the XMT scans (Figs. 2–7) a significant depth of the dental alveoli can be observed. The tooth crowns are sub-circular in the cross section, with only a marginally larger mesiodistal axis than the labiolingual one. In the maxillae and dentaries, they are attached to the bones at an average angle of 70° –in relation to other Teleosauroidea they can be described as non-procumbent (see Johnson et al., 2022a; Supplemental Material 3), but this varies along the lower and upper jaw, as the teeth of the dentary are slightly more angled in their position than the almost vertically oriented teeth of the maxilla. Note, however, that the dentary teeth are partly pulled out of their alveoli, so their inclination is not considered here fully representative.

The external surfaces of the tooth enamel are ornamented with apicobasal ridges. The ridging is densely spaced, and the relief of the ridges is intermediate in prominence (from 0.5 to 1 mm) according to Johnson et al. (2022a) classification. The ridging is denser on the lingual and distal surfaces of the teeth, than on the labial and mesial ones. When the ridges are excluded, the underlying enamel surface is smooth with no observable ornamentation in mid-crown section, but the apices exhibit faint dental ornamentation. This ornamentation, that is more clearly visible under magnification, can be described as an anastomosing pattern (see the closeups of teeth in Figs. 5F and 6F), which occurs as an extension of mid-crown ridging. Carinae, of low relief, are present on the mesial and distal surfaces that can be seen under magnification (see Fig. 5F, the carinae are more prominent on the erupting anterior tooth) and felt by hand examination. There is no evidence of serrations along the carinae. Most of the teeth are damaged, especially in the apical section, with broken/worn apices. Some teeth have their enamel layer damaged, exposing the inner dentine, and some have also exposed pulp cavities.

Principal coordinates analysis

The averaged dental parameters, which were used for the morphological assessment of dental function, consistently placed the Załęcze Wielkie specimen (MZ VIII Vr-72) within or in close proximity to the Machimosauridae morphospace. In PcoA analysis (Fig. 8), the Załęcze Wielkie specimen is located in a region of morphospace that is intermediate between the regions occupied by Teleosauridae (occupying the right side of the morphospace) and Machimosaurini (occupying the left side). In ordination space, the Załęcze Wielkie specimen is in the same region as non-machimosaurin machimosaurids (in particular the genera Deslongchampsina, Charitomenosuchus, and Neosteneosaurus). It should be noted that the Załęcze Wielkie specimen is especially close to one of the Neosteneosaurus edwardsi tooth crowns in morphospace.

Figure 8 Results of the teeth morphometry PCoA analysis based on averaged parameters of three complete and intact tooth crowns of the MZ VIII Vr-72.

Machimosaurini marked as yellow squares, Machimosauridae as red circles and Teleosauridae are green rhombuses while outgroup (Plagiophtalmosuchus gracilorostris) has grey, pentagonal signature. The highlighted spaces correspond to the machimosaurin, non-machimosaurin machimosaurids and teleosaurids. MZ VIII Vr-72 (black star) is localized on the margin of the Machimosauridae morphospace, closely adjacent to other macrophagous machimosaurids.

Discussion

The discovery of a preserved teleosauroid rostrum improves our understanding of the biodiversity of Jurassic vertebrates in south-central Poland. The presence of the Teleosauroidea in the region suggests that the Polish Jura shares similarities in vertebrate biota with the Submediterranean facies found in southern Germany and France, as well as the Subboreal facies of United Kingdom (Matyja & Wierzbowski, 1995; Foffa et al., 2019; Johnson, Young & Brusatte, 2020b). These areas are renowned for their rich record of the Teleosauroidea, unlike the Boreal facies of more northward regions like Spitsbergen which are rather known from occurrences of Plesiosauria and Ichthyosauria (Delsett et al., 2019). Teleosauroids, in turn, seem to have never colonized high latitude environments, which may correspond with their inferred lower basal body temperatures than other coeval marine reptile groups (Séon et al., 2020), being confined to warmer waters, in case of Europe of mainly Tethyan provenience, and this seems to be the case also for non-European occurrences (i.e. Buffetaut, Termier & Termier, 1981; Newton, 1893; Li, 1993; Fanti et al., 2016; Cortés et al., 2019; Dridi & Johnson, 2019). The sediments which later formed the Jurassic rocks of the modern-day Polish Jura are believed to have been deposited on the northern margin of the Tethys Ocean, belonging to the Submediterranean Province (Wierzbowski et al., 2016), suggesting that the Submediterranean belt provided conditions for these animals also more eastward.

The Teleosauroidea have a significant presence throughout Europe, and they are notably prominent through the whole Jurassic (see Johnson, Young & Brusatte, 2020b for a recent overview). Still, their fossil record is rather scarce in Poland, and with the loss of the valuable Czarnogłowy site due to the flooding (A. Wierzbowski, 2023, personal communication), highly important source for teleosauroid material, every occurrence is of importance. The fossil record of Thalattosuchia from southern Poland is even more lacking than the material from more northward, Subboreal (Wierzbowski et al., 2016) Pomerania, and thus the Załęcze Wielkie specimen (MZ VIII Vr-72) is a valuable addition that begins to fill this gap and extends the known geographic range of teleosauroids into the more eastward parts of Submediterranean province, which also covered current-day southern Poland.

The Załęcze Wielkie specimen within Teleosauroidea

In teleosauroid systematics, cranial morphology is extremely important, providing many of the diagnostic features (see examples of phylogenetic characters in Johnson, Young & Brusatte (2020b), and the diagnoses in Johnson et al. (2022b) and Young et al. (2024). The Załęcze Wielkie specimen only preserves the cranial rostrum, but its 3D preservation with intact, well-preserved teeth allows to draw some comparisons. The teleosauroids of the Oxfordian are still incompletely understood, especially when compared to the preceding Callovian fauna (e.g., Newton, 1895; Young et al., 2014a; Johnson et al., 2015; Foffa, Young & Brusatte, 2015). Most of the Oxfordian specimens have been initially identified as either ‘Steneosaurus’ or Machimosaurus. Some of them have been revised in Young et al. (2014a) and Johnson, Young & Brusatte (2020a, 2020b), but still some material requires revisions, and as such the study by Johnson, Young & Brusatte (2020b) is the most up to date source for comparisons.

We can place the Załęcze Wielkie specimen (MZ VIII Vr-72) in Teleosauroidea due to the following characteristics: (1) straightened (sub-rectangular) anterior maxilla in palatal view (differing from the tapering V-shaped terminal maxilla of the Metriorhynchidae); (2) the rostrum is largely formed by the maxilla, with no premaxilla-nasal contact, and the premaxilla contributes 25% or less of total rostral length; and (3) it being a thecodont rostrum with a relatively homodont dentition (Johnson, Young & Brusatte, 2020b).

We can exclude the Załęcze Wielkie specimen from Teleosauridae as it lacks the following apomorphies (Johnson, Young & Brusatte, 2020b; Johnson et al., 2022b): (1) the external naris is oriented either anteriorly or anterodorsally; and (2) the anterolateral premaxillary margins extend ventrally. The Machimosauridae are characterised as having: (1) a dorsally directed external naris; (2) a premaxilla lacking a ventral expansion (Johnson, Young & Brusatte, 2020b; Johnson et al., 2022b). Therefore, the morphology of the Załęcze Wielkie specimen is consistent with Machimosauridae.

Within Machimosauridae, the Załęcze Wielkie specimen preserves one of the apomorphies of the subclade Machimosaurinae (Johnson, Young & Brusatte, 2020b; Johnson et al., 2022b): non-procumbent dentition. However, we can exclude the Załęcze Wielkie specimen from subclade Machimosaurini as it lacks the following apomorphies (Johnson, Young & Brusatte, 2020b; Johnson et al., 2022b): (1) tooth crowns that have blunt and rounded apices; (2) apical tooth enamel ornamentation is composed of a strongly developed anastomosed pattern; (3) no curvature of the middle and posterior tooth crowns; (4) presence of false serrations particularly on the posterior dentition; and (5) presence of true denticles. Moreover, the tooth counts of members of Machimosaurini are generally reduced compared to other machimosaurines (e.g., Young et al., 2014a; Foffa, Young & Brusatte, 2015; Johnson, Young & Brusatte, 2020b), and although we cannot ascertain the exact tooth count of the Załęcze Wielkie specimen, its tooth count does not appear to be consistent with Machimosaurini.

Therefore, we propose that the Załęcze Wielkie specimen is a non-machimosaurin machimosaurine. With its the non-procumbent dentition and noticeable reception pits present along the entirety of the maxilla (most prominent in the anterior maxilla), the Załęcze Wielkie specimen most closely resembles Andrianavoay, Neosteneosaurus, and Proexochokefalos (Johnson, Young & Brusatte, 2020b). Andrianavoay is known from the Bathonian of Madagascar (Newton, 1893; Johnson, Young & Brusatte, 2020b), making it a less likely identification. The dentition of the Załęcze Wielkie specimen does differ from those of Neosteneosaurus and Proexochokefalos, by having a slight apical anastomosed pattern (although not as strongly developed as in Machimosaurini). The dental ornamentation in Proexochokefalos is less prominent than in Neosteneosaurus (Johnson, Young & Brusatte, 2020a), which makes MZ VIII Vr-72 more similar to Neosteneosaurus. Similarly, the premaxilla is only moderately constricted and appears to be horizontally straight (besides deformation) in MZ VIII Vr-72, which is also more similar to Neosteneosarus than to Proexochokefalos (Johnson, Young & Brusatte, 2020a). Both Neosteneosaurus and Proexochokefalos lack an apical shift in enamel ornamentation (Johnson et al., 2022a). Unfortunately, there is not enough of the Załęcze Wielkie specimen preserved for us to conclusively determine if it belongs to either of those genera, or if it is a new taxon. As such, we consider it to be Machimosaurinae gen.et sp. indet.

The importance of the Załęcze Wielkie specimen

The Załęcze Wielkie specimen is most similar morphologically to the genera Neosteneosaurus (=‘Steneosaurus’ edwardsi) or Proexochokefalos (=‘Steneosaurus’ heberti, see above), with more similarities shared with the former. Both genera are known from Western Europe during the Callovian, with Proexochokefalos continuing to be present into the early Kimmeridgian (Johnson, Young & Brusatte, 2020b). Both Neosteneosaurus and Proexochokefalos were large-bodied machimosaurines with a mesorostrine cranial morphology, robust dentition and enlarged adductor musculature (Vignaud, 1997; Foffa et al., 2018; Johnson, Young & Brusatte, 2020b; Johnson et al., 2022a). While it is possible that the Załęcze Wielkie specimen is a representative of an unknown taxon, it does appear to have been one of the macrophagous machimosaurines that lived in Europe during the Middle and Late Jurassic. Outside of Europe from this time interval, teleosauroids are known from the Callovian of India and Tunisia (see Phansalkar, Sudha & Khadkikar, 1994; Khadkikar & Phansalkar, 1995; Dridi & Johnson, 2019). Unfortunately, these specimens are either fragmentary or poorly preserved, precluding a definitive identification.

The teleosauroid fauna of Western Europe dramatically restructured during the Late Jurassic. During the Callovian there was a diverse fauna of teleosauroids, dominated by machimosaurids, with longirostrine and mesorostrine forms present (e.g., Hua & Buffetaut, 1997; Vignaud, 1997; Foffa et al., 2018; Johnson, Young & Brusatte, 2020b; Johnson et al., 2022a). However, during the Oxfordian there appears to be the loss of longirostrine machimosaurids, and the dominance of machimosaurin machimosaurids (Foffa et al., 2018). During the Kimmeridgian, there were three groups of teleosauroids in Western Europe: (1) the durophagous Machimosaurini (i.e., the genus Machimosaurus), (2) the longirostrine aeolodontin teleosaurids (such as Bathysuchus), and (3) the mesorostrine machimosaurine Proexochokefalos (Hua & Buffetaut, 1997; Vignaud, 1997; Foffa et al., 2018, 2019; Johnson, Young & Brusatte, 2020b). By the Tithonian, only Machimosaurini and Aeolodontini are known to be present in Western Europe (Johnson, Young & Brusatte, 2020b).

Therefore, we seem to be seeing the extinction of first longirostrine machimosaurids, then non-machimosaurin macrophagous machimosaurids, in Western Europe during the Oxfordian–Kimmeridgian. In this light, the Załęcze Wielkie specimen tells us that non-machimosaurin macrophagous machimosaurids were present in Poland during this time interval. Although we cannot be sure, the Załęcze Wielkie specimen does not appear to be Proexochokefalos, suggesting that other non-machimosaurin macrophagous machimosaurids were present in the Late Jurassic. The ornamentation of the enamel on the apices, in the form of anastomosing pattern, is also an interesting feature of MZ VIII Vr-72, as the presence of this feature on the apices of the teeth has been described as present only in Machimosaurini (Johnson, Young & Brusatte, 2020b).

Palaeobiological implications inferred from observed morphology

Based on the comparison with suspected closely related genera, it is possible that the total length of the complete cranium of MZ VIII Vr-72 before the flexure of the rostral end could fit within the range of circa 80 cm in total (Fig. 9), but this estimation depends on relative proportions of the posterior cranium to the rostrum. Additionally, retroarticular processes of the mandible would further extend total skull length posteriorly to about 90 cm.

Figure 9 Overview of the structure of the rostrum (MZ VIII Vr-72) with interpretative drawings.

The CT scans (A, C, D, F, G–J, M, N, Q, R) and schematic drawings (B, E, K, L, O, P) in lateral, dorsal, ventral and frontal view with internal cross-sections along with reconstruction of the cranium (S) based on the non-machimosaurin machimosaurids (without the in-vivo deflection). The dorsal view of horizontal cross-section of nasal cavity highlights the almost 90 degree deflection of the anteriormost rostrum. Areas with hatched, translucent lines correspond to damaged or broken bone. Abbreviations: mx, maxilla; dent, dentary; pmx, premaxilla; ns, nasal.

The midline suture between the maxillae is well visible in dorsal view, yet narrow, and has observable lamellae in posterior part. The fragment of the midline suture of the dentary can likewise be observed in the natural longitudinal section of the rostrum along the surface of the block of rock matrix. In contrast, in ventral view, the midline suture of the maxillae is hardly visible. The observed pattern leads us to believe that the specimen described was not yet fully mature, as we would expect the suturing to create a visible distance between the maxillae in more ontogenically mature animals. This observation is based on modern crocodylomorph skull ontogenetic features, since in the older, mature specimens, sutures are more prominent than in younger ones (Bailleul et al., 2016). It is possible that a similar skull development pattern existed in teleosauroids.

The comparatively robust tooth crowns, together with worn apices of the Załęcze Wielkie specimen dentition suggest that the specimen was not limited to feeding on soft-bodied prey. Our PCoA analysis further supports our contention that the Załęcze Wielkie specimen had an intermediate dentition morphology between the macropredatory/durophagous genera such as Machimosaurus and the obligate piscivores like Teleosaurus (albeit the Załęcze Wielkie specimen shares more similarities with the former), as it was recovered alongside other Machimosaurinae such as Neosteneosaurus. The dentition of the Załęcze Wielkie specimen fits into the ‘B1 morphotype’ outlined by Vignaud (1997). During the Callovian-Oxfordian, the ‘B1 morphotype’ was found in the species Steneosaurus edwardsi, S. durobrivensis and S. heberti (Vignaud, 1997), these species are now referred to the genera Neosteneosaurus and Proexochokefalos (Johnson, Young & Brusatte, 2020b). Vignaud (1997) defined the ‘B1 morphotype’ as having regular apicobasal enamel ridges that can anastomose in the apical region, an average apex (neither pointed nor rounded), and a height-to-length ratio between 2.0–2.3. The Załęcze Wielkie specimen shares these features, and its crown height/width ratio is 2.17. Vignaud (1997: 55) considered the species that had the ‘B1morphotype’ to be both mesorostrine and opportunistic feeders, citing Mazin (1988) in putting forward a prey envelope consisting of cephalopods, fish, and reptiles.

Massare (1987) placed Neosteneosaurus edwardsi (referred to as ‘Steneosaurus’ durobrivensis therein) in her ‘pierce II guild’, which she inferred had a preference for fish, and did not specialise in either soft or bony prey items. Foffa et al. (2018) quantified the Massare (1987) dental guild system, and they placed Neosteneosaurus edwardsi (referred to as ‘Steneosaurus’ durobrivensis and ‘S.’ edwardsi therein) and Proexochokefalos heberti (herein referred to as ‘S.’ heberti) in their ‘Pierce guild’. They noted that extant animals with similar dentition to their ‘Pierce guild’ “are inferred to be fish and squid eaters” (Foffa et al., 2018: 1549). As with the Massare (1987) qualitative dental guild system, the Foffa et al. (2018) classification places machimosaurids in a region of morphospace that is not specialised for either soft-bodied prey or hard-bodied prey items.

The Załęcze Wielkie specimen also seems to fall into the ‘robust’ jaw type described by Johnson et al. (2022a), as it shares the same general morphology as machimosaurids. Consequently, it can be characterized as mesorostrine, according to the ecotypic classification proposed by Johnson, Young & Brusatte (2020b) and Johnson et al. (2022a) (and the dental classification by Vignaud, 1997). This morphology is generally consistent with the inferred placement in Machimosaurinae. The Machimosaurinae have been labelled as semi-marine generalists (Johnson, Young & Brusatte, 2020b), and their mesorostrine condition together with robust dentition are congruent with this niche.

In coronal view, the rostrum (Fig. 7) is divided into three large cavities: ventrally for the nasal cavity, and dorsally for the dorsal alveolar canals. In extant brevirostrine crocodylians, the dorsal alveolar canals are not positioned as medially, and in those taxa, there is a series of internal cavities for the paranasal sinuses (Witmer & Ridgely, 2008; Bourke, Fontenot & Holliday, 2022; Cowgill et al., 2022). The morphology of the Załęcze Wielkie specimen is far more similar to that observed in longirostrine thalattosuchians and the extant Indian gharial (Pierce, Williams & Benson, 2017; Bourke, Fontenot & Holliday, 2022; Bowman et al., 2022). Note, the extant Indian gharial has a pneumatised rostrum, but the sinuses largely invade the posterior maxilla (see Cowgill et al., 2022). It is likely that the nasal chamber was separated by the central septum, not unlike in the gharials (Bourke, Fontenot & Holliday, 2022, fig. 16), but in the fossil this feature is not preserved.

Rostral pathology

The Załęcze Wielkie specimen (MZ VIII Vr-72) has a strongly inflected anterior rostrum (Figs. 3, 4, 7, 9). In the CT scans, its internal structure shows only slight buttressing at the right maxilla, yet with the visible asymmetry in the size of the tooth alveoli, with the alveoli of the right maxilla being smaller than those from the left maxilla. The premaxilla and the anterior-most maxilla, along with the nasal cavity, are at an angle of about 90 degrees compared to the rest of the rostrum. The preserved part of the dentaries shows no abnormal flexure, despite being preserved in close association and probable anatomical position, indicating that only the upper jaw was affected. Based on these observations, we suggest that this observed condition was present in-vivo (Fig. 10) and is most likely not a result of diagenetic processes. The misalignment of the jaws is a common occurrence in modern-day crocodylomorphs (e.g. Webb & Messel, 1977; Webb, Manolis & Buckworth, 1983; Montague, 1984), and jaw pathologies have also been observed in machimosaurids (Hua, 1996). This pathological development could have strongly affected the life of the specimen. Nonetheless, the pathology appears to be fully healed and influence the alveoli and tooth size and layout, indicating that the individual survived and was able to develop new generations of teeth. Therefore, if it resulted from trauma, it apparently did not prevent it from acquisition of food over an extended period of time.

Conclusions

Herein, we redescribed the Załęcze Wielkie specimen (MZ VIII Vr-72), the rostrum of a teleosauroid crocodylomorph from southern-central Poland. It has a pronounced lateral deflection of the anterior rostrum, which is interpreted as in vivo deformation, and is an example of the misalignment of the crocodylomorph jaw apparatus. Based on our comparisons with other teleosauroids, the Załęcze Wielkie specimen can be referred to Machimosaurinae, and appears to be most similar to the genera Neosteneosaurus and Proexochokefalos. Although, we cannot preclude the possibility that it is a new taxon (due to the poorly developed anastomosed ornamentation patterns of the tooth crowns), we can exclude the Załęcze Wielkie specimen from the machimosaurine subclade Machimosaurini, as it lacks several of their autapomorphies (including lack of crown curvature, blunt tooth apices and presence of tooth serrations).

Our ordination analyses support our comparative anatomical assessment. The Załęcze Wielkie specimen is consistently found to be in the non-machimosaurin machimosaurid region of morphospace, close to the genus Neosteneosaurus. This suggests that the specimen was a macrophagous machimosaurid, like Neosteneosaurus and Proexochokefalos. Macrophagous non-machimosaurin machimosaurids appear to have become extinct in Western Europe during the Kimmeridgian, with the machimosaurins becoming the only remaining macrophagous teleosauroids. The Załęcze Wielkie specimen shows that outside of Western Europe, macrophagous non-machimosaurin machimosaurids were present during the Oxfordian. Future discoveries will be needed to determine whether the machimosaurid fauna of Eastern and Central-Eastern Europe followed the same trends as those in Western Europe. The eurocentrism in the study of thalattosuchians hampers our understanding of the group (Young et al., 2024). But there is also a great discrepancy in our knowledge of thalattosuchians from across Europe, with the Central-Eastern and Eastern European fossil record largely understudied. This means we do not understand known species full geographic ranges, how provincial the European faunas were, and whether there were localized extinctions through time. We hope that our re-description of the Załęcze Wielkie specimen will spur future workers to study the thalattosuchians of Central-Eastern Europe.

Supplemental Information

Supplemental Information 1 Discrete and continuous characters used in PCoA analysis.

Supplemental Information 2 Modified dataset of teleosauroid teeth characters from Johnson et al. (2022a) including MZ VIII Vr-72 data.

Supplemental Information 3 Results of the PCoA analysis, including eigenvalues and other coordinates.

We would like to thank Stanisław Kugler (Institute of Geological Sciences, Jagiellonian University) for preparing the paleoart of the studied specimen, which is represented at Fig. 10 and Prof. emeritus Andrzej Wierzbowski (University of Warsaw, Faculty of Geology) for consultations concerning the aspects of local geology of Załęcze Wielkie. Thanks are extended towards Katarzyna Przestrzelska, who helped with preparation of the specimen. Gratitude is expressed towards the Editor Jérémy Anquetin and Reviewers (Stéphane Hua and anonymous Reviewer) for their insightful comments concerning the manuscript.

Additional Information and Declarations

Competing Interests

Author Contributions

Data Availability

Mark Young is an Academic Editor for PeerJ.

Łukasz Weryński conceived and designed the experiments, performed the experiments, analyzed the data, prepared figures and/or tables, authored or reviewed drafts of the article, and approved the final draft.

Błazej Błażejowski conceived and designed the experiments, performed the experiments, prepared figures and/or tables, authored or reviewed drafts of the article, and approved the final draft.

Tomasz Szczygielski conceived and designed the experiments, performed the experiments, analyzed the data, prepared figures and/or tables, authored or reviewed drafts of the article, and approved the final draft.

Mark T Young analyzed the data, authored or reviewed drafts of the article, and approved the final draft.

The following information was supplied regarding data availability:

The measurements and observations of teleosauroid teeth morphologies, based on Johnson et al. (2022a) used in PCoA analysis, are available in the Supplemental Files.

The virtual scans of specimen MZ VIII Vr-72 rostrum are available at MorphoSource:

- The F1 section: https://doi.org/10.17602/M2/M609339 (CT images series) https://doi.org/10.17602/M2/M600506 (3D mesh);

- The F2 section: https://doi.org/10.17602/M2/M609720 (CT images series) https://doi.org/10.17602/M2/M600521 (3D mesh);

- The F3 section: (CT images series) https://doi.org/10.17602/M2/M609868 (CT image series); https://doi.org/10.17602/M2/M600532 (3D mesh);

- Full composite of F1, F2 and F3 3D meshes: https://doi.org/10.17602/M2/M609361.

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
