# Peer review of "The first occurrence of machimosaurid crocodylomorphs from the Oxfordian of south-central Poland provides new insights into the distribution of macrophagous teleosauroids"

_PeerJ, doi:10.7717/peerj.17153_

## Round 0.1 · original submission · Minor Revisions

Your manuscript was reviewed by two external reviewers and myself. We all agree that this work is interesting, but we have a number of comments that we would like you to take into consideration in revising your text. Please disregard the occasional harsh tone of the first reviewer to concentrate on the essential. The two reviewers notably point some lacking in bibliography and morphological comparison. Please provide a detailed reply to the reviewers with your revised version.

Here are a few additional comments that I would like you to address.
- For reader convenience, I would like to see all institutional abbreviations (incl. those in Table 1) grouped in a specific section in the manuscript (same for the supplementary files).
- The Morphosource links are not provided, which precluded examination of the scans by the reviewers and myself. I am also interested to know if the 3D models and the raw CT data (image stack) will be made available to readers. In my opinion, this is really important that all data be made openly available. Also the links should be indicated in the body of the manuscript, not just in the supplementary files.
- lines 145 and 209: the preservation of the specimen in several parts is too confusingly explained. F1, F2, and F3 are mentioned out of the blue line 145, then 'three consecutive fragments' (supposedly F1, F2, and F3) are mentioned line 209. Please review your description from the perspective of the reader.
- lines 149–150: please clarify these 'most effective methods for digital processing and analysis of tomographic data'
- line 153: it is common to list the scans parameters somewhere.
- lines 179–181: not sure I understand the purpose of this section


Minor comments:
- line 70: 'or to the "Steneosaurus"' either something is missing or you need to delete 'the'
- line 142: duplication of 'of the'
- line 515: it seems you inverted Neosteneosaurus and Steneosaurus (inconsistency with lines 505–508)
- line 519: check usage of Steneosaurus and 'Steneosaurus'

·

Basic reporting

Great weakness in biblography : nearly only recent works and only from coworkers.
It lacks some references that could have helped the authors. This paper is flooded by the smoggy classification use by coworkers

Experimental design

it is like the authors are hidden behind a sofware and don't use a classical description : a sofware that you haven't developped, you are not sure at 100% of the rsult, if the result is correct and obsolete in 5 years max. Machimosaurus is based on a well known tooth morphology : why you dont use it? as a result you are unable to determine your material

Validity of the findings

unable to determine the species so to compare with other discoveries is speculative; Moreover due to you r incomplete bibliography, you miss some things like the already known pathology of Machimosaurus mandible or their heterodonty illustrated by M. Mosae.
Europocentrisme of discovery : of course yes when you read only some authors

Additional comments

28 why semi aquatic ? not found in pure terrestrial deposits
32 Teleosauroid = Teleosaurida, precise which classification and avoid the actual smoggy one
35 non-machimosaurin machimosaurid : does it mean something ? please use a real classification and precise which one
37 you could compare teeth only with softwares ? do you exactly how it runs ? if it has bugs ? please you are a pallaontologist, use your brain
39 : wonderful, size variation, age, sexe there is no evolution or difference ? see M. mosae mandible heterodonty
51 please respect the early workers like Martill
56 complete you bibliogrpahy and cite at least Vignaud or Hua, you have a gap of 30y where many papers have been published
63 Fanti was not the first to tell it, you really read Buffetaut 1982 ?
65 stop «non-machimosaurin machimosaurid « this is not because some authors like to create taxa from one specimen : this is a machimosaurid or not ? this is exactly if you say a non human hominid so an ape ? this is the same case. This is why i hate the actual works the don’t respect the international code and its stability

66 aeolodontin teleosaurids and machimosaurin machimosaurids so teleosaurids ?

123 a teleosaurid is a crocodylomorph…
158 please respect early workers at least MaSSare 1987 , Massare, J.A. Tooth morphology and prey preference of Mesozoic marine reptiles. Journal of Vertebrate Paleontology 1987, 7, 121-137.
not only your co workers…..34 years ago on all forms of teeth or Vignaud 1995 : Vignaud, P. La morphologie dentaire des Thalattosuchia (Crocodylia, Mesosuchia). Paleaeovertebrata 1997, 26, 1-4, 35-39.

177 you don’t have eyes ? really, complete first your bibliography instead of using a sofware you haven’t code, you can’t predict the results of such softwares or limitation. Like all sofwares, in 5 years max obsolete, and your results too ?

181 using the ICZN ? with such sentence of non-machimosaurin machimosaurid ?
191 The base of the classification is the species : so you are unable to determine the species the genus but able to tell which subfamily ???? you are kidding ?

225 and what about french and german specimens ?
253 really complete your bibliography nothing between 1913 and 2020 ???? any comparisons with french specimens ? you are speaking of Machimosaurs !!!!! not even a comparison with the M. Mosae of Boulogne sur mer or the hugii of Leiria ?

257 which genera ? you don’t even speak of complete skeletons found in the rest of Europe.
287 as usual your bibliography is weak for pneumacity see at least Hua, S.; De Buffrenil, V. Histology of the Thalattosuchia as a clue of the interpretation of functional adaptations in the Thalattosuchian (Reptilia, Crocodylia). Journal of Vertebrate Paleontology 1996, 16, 4, 703-717

328 complete at least your bibliography seriously like Vignaud 1995 before citing your
Coworkers : have you check also the work of Mueller Towe of on Teleosaurids of Holzmaden?

329 so in contradiction with your previous sentence (please check my monography on the mosae, I have illustrated the heterodonty inside the same mandibleof a Machimosaurus…

344 please check Massare and vignaud works… See Krebs 1968 the best illustrations of Machimosaurus teeth, check my paper of my Machimosaurus found in Veller sur mer… really complete your bibliography

358 in a previous sentence you said it depends of the position and what about the other in this fanstastic software : all positions were well referenced? If not garbage

364 please for such genera ask to your coworkers, they are future nomen dubium don’t use it (iczn reports in progress)
Hua, S.; Pennetier; E. Pennetier, G. A juvenile Steneosaurus in the Callovian of Normandy (France); a genus too hastily consigned to the wastebasket? Carnets natures 2021, 8, 1 -8.

In Fact for all other durophageaus species of steneosaurus (again we see the limit of a specimen = a new genera)

And what about your idea ? no idea except from the software ? what is your add value ?

371 you dont even compare to neighboroood discoveris only English…
385 of course sentence coworkers they are going to tell of the proeminnce of crocodilians they are working exclusively on it please check a real synthesis like Martill for the oxford clay

395 please make seriously your bibliography before such sentence and respect previous works made by workers not only authors working on only 2 families without knowledge of the other groups
399 so in oxford clay nothing between 1895 and your coworkers ?
402 please check machimosaurin machimosaurids = machimosaurus ?
Hua, S.; Pennetier; E. Pennetier, G. A juvenile Steneosaurus in the Callovian of Normandy (France); a genus too hastily consigned to the wastebasket? Carnets natures 2021, 8, 1 -8.
To see which article of the ICZN has not been respected by this team for Steneosaurus
Complete your Bibliography !!!!!

418 a complete paragraph without comparing with the definition of Machimosaurus by its teeth (again see Krebs 1968) I resume one page : If your teeth is not the same this is the holotype of the genus Machimosaurus this is not a Machimosaurus and nothing else!!! This is really basic paleontology, instead of being floaded in papers
You use only your coworkers works, and often full of round cycle demonstration, please check what has been made beside and before!!!! In paleontology you have not just the use the last papers of your coworkers found in internet

448 a complete paragraph of comparisons with your teeth that you are unable to say which species ? speculative,
463 machimosaurin machimosaurids = Machimosaurus ! please such expressions are ridiculous

466 to 467 please use Steneosaurus again see Hua, S.; Pennetier; E. Pennetier, G. A juvenile Steneosaurus in the Callovian of Normandy (France); a genus too hastily consigned to the wastebasket? Carnets natures 2021, 8, 1 -8.
Even your coworker agree now that Steneosaurus is valid

470 this is not you have not found fossils that they are not existing, if you have not the good deposit, again a speculation

482 you don’t even compare to the only known skeleton of Machimosaurus skeleton, great job… what is the interest of this paragraph : you are not been able to determine this specime ? compare when you could compare

545 Rostral pathology : if you have done correctly your bibliography you will have seen the the M. mosae of Boulogne sur mer has pathology but worse for you the pathology of Machimosaure mandible is already known and described :
HUA S. 1996 – rééexamen du Machimosaurus cgf. Hugii des carrieres d’Haudainville (Meuse, Est de la France) : contribution à l’étude du genre Machimosaurus MEYER, 1838. Bulletin de la société géologique de Normandie et des amis du Museum du havre, 83, 1 & 2, p11-16

This specimen of Machimosaurus had the anterior part broken, and the occlusion modified in Vivo
Again it underlines your weakness in bibliography

562 First paragraph you compare your unknown species to future nomen dubium , so without interest
583 No !!! Eurocentrism just for your coworkers : check Zulma Gasparini, Jeremy Martin, Buffetau works and even Bardet and me (1996) about Simolestes nowackianus, in fact a Machimosaurus.
Your incomplete bibliography makes you tell you inacurrency things

Reviewer 2 ·

Basic reporting

I had the opportunity to review the above-mentioned work, which documents new occurrences of teleosauroid thalattosuchians from an under-represented region. Although the material is incomplete, this study demonstrates that it is sufficiently diagnostic to recognize its referral to machimosaurids. This is important because most previous records come from Western Europe, and complete skull specimens are especially rare. As indicated by the authors, the Polish discoveries comforts the distribution of machimosaurids to the northern margin of the Tethys.

The study is generally well conducted and I particularly appreciated the quality of the morphological investigation using CT scan data, that help complementing external observations.

My comments are relatively minor and the suggestions below should be easily answered/fixed:

Line 60: you can mention that thalattosuchians never colonized high latitude environments.
Line 63: the presence of teleosauroids in the Cretaceous has been questioned by several authors (e.g. Martin et al., 2019, but not only). I suggest also to mention a possible teleosauroid occurrence from Colombia here (Cortés et al. 2019) and rephrase your sentence accordingly.

Line 67; the whole paragraph requires a citation (e.g. Foffa et al. 2018)

Line 75: citing a map could be useful

Line 80: what is the age of these various occurrences? Are they relevant to the Oxfordian specimen reported in the present study. Please specify and re-arrange the paragraph accordingly.

Line 371: “significantly” can be deleted to avoid subjective overstatement.

Line 378: again, thermometabolism may explain the absence of thalattosuchians in high latitude environments. You should be able to cite a few references on this matter.

Line 382: delete “hosted”

Line 383: delete “the” in front of “Europe”

Line 384: again, the Early Cretaceous occurrences should be taken with caution

Line 387: please rephrase

Line 391: admninistrative borders are not relevant, therefore I'd suggest to rewrite this part in relation to paleogeographic or geological domains, similarly to the beginning of the paragraph.

Line 394: yes, but this generality applies to many tetrapod groups. One major limitation with teleosauroids, however, is the convergence in longirostry that renders cranial characters difficult to interpret. The recent review papers cited here unfortunately do not take into account such a problem and tend to erect a high number of genera and species.

Line 448, Title: please indicate why the specimen is of importance

Line 449: The two newly mentioned genera (Neosteneosaurus and Proexochokefalos) were erected recently. It would be worth re-mentionning their provenances and ages to help the reader follow the recent taxonomic changes.

Line 455: are there any lines of evidence to discuss the possibility of the new specimen to be a new species?

Line 462: this pattern (loss of longirostrine forms) may be largely biased by the nature of the fossil record.

Line 464: about point (1) and the genus Machimosaurus, several previous studies are lacking and should be cited here, especially as concerns the most complete specimens (Sauvage and Liénard, 1879; Buffetaut, 1982; Hua et al., 1993; Martin and Vincent, 2013; Martin et al., 2015; Young et al., 2015).

Line 476: I think it is premature to draw a conclusion about the pattern of enamel ornamentation given the patchy fossil record and poorly resolved relationships within Teleosauroidea.

As a general comment, this contribution lacks a comparison with other specimens from other parts of the Tethys and more or less of the same age such as teleosauroid remains described from the Middle Jurassic of Tunisia (Didri and Johnson, 2019 Geobios). The biogeographic occurrence should be included in the discussion as well and would probably support the conclusions of the present study on the implications for thalattosuchian biodiversity outside of the Western Tethys.

Experimental design

please refer above

Validity of the findings

please refer above

Additional comments

please refer above

---

## Round 0.2 · accepted · Accept

With this revised version, I confirm that you have taken the comments of the two reviewers and myself into consideration. Thanks for your patience in replying to the comments of the first reviewer. Thanks also for the clarification of the MorphoSource links. However, I see that you share only the 3D meshes (.ply files) you created from the CT scan, but not the original TIFF stack. In my opinion, it is important that you share also the TIFF stack so that others can reproduce your results and expand on your data. You can easily add the stack to your MorphoSource project and add the link in the manuscript at proof stage.

The Section Editor agrees that the image stack should also be uploaded to MorphoSource.